# Comparing Approaches for Reconstructing Groundwater Levels in the Mountainous Regions of Interior British Columbia, Canada, Using Tree Ring Widths

**Stephanie C. Hunter [1],***, **Diana M. Allen [1]** and **Karen E. Kohfeld [2]**

[1]    Department of Earth Sciences, Simon Fraser University, Burnaby, BC V5A 1S6, Canada; dallen@sfu.ca
[2]    School of Resource and Environmental Management, Simon Fraser University,
       Burnaby, BC V5A 1S6, Canada; kohfeld@sfu.ca
*    Correspondence: stephanie_hunter@sfu.ca

**Abstract:** Observed groundwater level records are relatively short (<100 years), limiting long-term studies of groundwater variability that could provide valuable insight into climate change effects. This study uses tree ring data from the International Tree Ring Database (ITRDB) and groundwater level data from 22 provincial observation wells to evaluate different approaches for reconstructing groundwater levels from tree ring widths in the mountainous southern interior of British Columbia, Canada. The twenty-eight reconstruction models consider the selection of observation wells (e.g., regional average groundwater level vs. wells classified by recharge mechanism) and the search area for potential tree ring records (climate footprint vs. North American Ecoregions). Results show that if the climate footprint is used, reconstructions are statistically valid if the wells are grouped according to recharge mechanism, with streamflow-driven and high-elevation recharge-driven wells (both snowmelt-dominated) producing valid models. Of all the ecoregions considered, only the Coast Mountain Ecoregion models are statistically valid for both the regional average groundwater level and high-elevation recharge-driven systems. No model is statistically valid for low-elevation recharge-driven systems (rainfall-dominated). The longest models extend the groundwater level record to the year 1500, with the highest confidence in the later portions of the reconstructions going back to the year 1800.

**Keywords:** mountain hydrogeology; tree rings; paleoclimate

## 1. Introduction

Mountains are a key supplier of water for many regions around the world, producing invaluable freshwater resources for both human use and natural ecosystems. A global assessment of the significance of mountains for water resources showed that 44% of global mountain areas played an important or essential role in the local water supply [1]. However, these mountain water resources are also some of the most vulnerable to climate change [2,3]. Recent trends in water availability in mountain regions strongly suggest that the seasonality of mountain runoff has changed due to decreasing winter snowpack, leading to changes in water availability throughout the year [4].

Groundwater is a particularly important resource in mountain regions, as it is the primary source of water for many streams during periods of low streamflow. Therefore, it is important to understand how climate variability affects the groundwater system. However, the effects of climate change on groundwater in mountain areas remains an area that is understudied [5]. In particular, the response of groundwater to climate variability over time is necessary to put current observations into a long-term

perspective, as the observed records of groundwater may not be representative of all of the conditions that are possible in a groundwater system [6]. Our current observations of hydroclimate variability cannot show the full range of natural variability, as many records only extend as far back as 100 years at most. Previous paleoclimate studies suggest that North America likely experienced droughts of greater severity and magnitude than those that occurred during the observational record, and that the period of record of many observational records began during an unusually stable climatic period compared to the paleoclimate record [7,8]. Creating paleoclimate reconstructions of groundwater levels can extend these records and provide new insights into an important component of the natural hydrologic cycle.

The width of tree rings provides annual measurements of tree growth, which occurs in response to climatic factors such as temperature, precipitation, snowpack, and soil moisture. In the Pacific Northwest, tree species that are moisture sensitive may respond to rain or snowfall depending on elevation [9]. At high elevations, winter snowpack may play a role in tree growth, as the timing and amount of snowmelt dictates the length of the growing season and the width of annual tree rings [9–11]. At lower elevations, rainfall may play more of a role in determining the moisture supply and tree growth [9]. These climate variables also affect groundwater levels in mountain regions, with increased snowpack and rainfall, both leading to increased groundwater levels. Moreover, while the main source of water for trees is soil water, there is evidence that some trees may be able to access groundwater, in particular during the dry season [10]. Therefore, tree rings are a potential candidate for reconstructing past groundwater levels.

Tree rings are commonly used to reconstruct streamflow e.g., [12–24] but only rarely have been used to reconstruct groundwater levels. Perez-Valdivia and Sauchyn [25] used five tree ring records from the Rocky Mountain foothills to reconstruct groundwater levels at two observation wells in the Interior Plains Region of Alberta, Canada. They were able to extend groundwater level records by 300 years and identified longer drought periods in the past groundwater levels records than had been observed in historical groundwater levels records. Ferguson and St. George [26] used tree ring, temperature, and precipitation records from a riparian study site to reconstruct average groundwater levels of shallow wells near in Manitoba, Canada, back to the year 1907. Although the reconstruction was short, they found that past groundwater levels had increased variability compared to observed groundwater levels, particularly in the time period from 1930 to 1942. Tree ring records have also been used in combination with other modeling techniques; tree rings were used along with gravimeter observations to reconstruct groundwater storage in Germany [27] and as part of an artificial neural network model to predict groundwater levels in alluvial aquifers in Iran [28,29]. However, these studies were undertaken in topographically flat areas where large regional aquifers exist, such as the Canadian Prairies, and in areas where shallow water tables may be the main source of water for riparian trees. In mountainous regions, groundwater systems are much more complex, with strong groundwater–stream connectivity owing to the high relief. Thus, mountain regions offer a unique setting for reconstructing groundwater levels.

Reconstructing groundwater levels in mountainous regions introduces complexities into the approach for several reasons. First, tree ring records are often located high in the mountains, while the observation wells in which the groundwater levels are recorded are often located in valley bottoms. Second, observation wells are at different locations in an aquifer, some in proximity to streams and others not, and so are variably influenced by aquifer–stream connections and diffuse recharge. Third, the aquifer materials and the associated hydraulic properties which control the groundwater level response, are diverse in valley aquifer systems, so two wells in the same climate regime but in different aquifers may respond differently to climate. Finally, streams may be sourced in the high mountain areas and therefore are driven by snowmelt in the spring and summer months. When the streams enter the valley bottom, they can recharge the aquifers at lower elevations.

Allen et al. [30] examined observation well records from different hydroclimatic regions in British Columbia (BC) to reconcile the varied groundwater level responses in different settings. They defined and classified the aquifer–stream system type as recharge-driven or streamflow-driven based on

the response of the groundwater level in comparison to streamflow. In a recharge-driven system, the groundwater level response leads the streamflow response throughout the year, and diffuse recharge to the aquifer continuously provides baseflow to the stream. In a streamflow-driven system, the streamflow leads the groundwater level response. For example, during the spring freshet, stream level is higher than the groundwater level and the stream recharges the aquifer causing the groundwater levels to rise. When the streamflow falls, the groundwater flow responds by reversing direction and discharging into the stream. Allen et al. [30] described these two recharge mechanisms as end members and suggested that other factors (e.g., distance to the stream or hydraulic connectivity with the stream) may result in less distinct responses. In this paper, each aquifer–stream system type is associated with a dominant recharge mechanism for the aquifer (streamflow-driven or recharge-driven). These recharge mechanisms may have important implications for groundwater level reconstructions, as the groundwater levels in each regime may be related to tree growth through differing mechanisms, or at different times of the year.

Another challenge involves defining the region in which tree ring sites can be expected to have a process-oriented relationship with groundwater levels in a given aquifer, and therefore be used to reconstruct groundwater level response. Some studies have used tree ring chronologies located in the same watershed, in close proximity to the hydroclimate records being reconstructed e.g., [23–25]. Cook et al. [31] used records within a 450-km search radius, rationalizing this wider search area because tree rings often respond to large-scale climate dynamics. This approach allows for information from more tree rings' chronologies to be used, which may be particularly important when few tree ring records are available nearby. More recent studies have used a wider area to select tree ring records by defining a "climate footprint", which is an area defined by a spatial correlation between the climate variable of interest and a climate index [13,20]. This climate footprint is meant to define an area where trees would be expected to respond similarly, thus providing a more climate-based rationale for selecting tree ring records. For hydroclimate variables, drought indices such as the Palmer Drought Severity Index (PDSI) and Standardized Precipitation Evaporation Index (SPEI) are ideal because they include multiple climate variables that are important to groundwater levels (temperature, precipitation, evaporation), and therefore would identify an area of similar response to climate conditions. However, although tree growth is often considered to be correlated to regional-scale climate conditions [31], climatic regimes can change over relatively small spatial scales in mountainous areas due to the rapidly changing topography [3]. Therefore, it is possible that this approach may not be suitable for mountain areas.

The goal of this study is to determine appropriate methods for reconstructing groundwater levels using tree ring widths in a mountainous setting. The study area is the southern Interior Plateau Region of British Columbia (BC), Canada. Tree ring records (available from the International Tree Ring Database, ITRDB [32]) from western North America that cover the period from 1900 to 2004 (or longer) are used to model past groundwater levels (collected from the BC Provincial Observation Well Network [33]) using combinations of (1) different recharge mechanisms and (2) different regions to select tree ring records. First, the study considers the dominant recharge mechanism by comparing the skill of the reconstruction using groundwater records that have been differentiated according to aquifer–stream system type to that of the average of all groundwater records. This approach is similar to that used for reconstructing streamflow, as the average of streamflow gauges is often used e.g., [17,34]. Second, the region from which tree ring records are selected for the reconstruction model is evaluated. Two types of climate-defined regions were used to select tree ring records: the climate footprint approach (described above), and various North American Ecoregions, which classify regions based on the dominant vegetation, soils, topography, and climate [35].

The southern Interior Plateau Region of BC (henceforth referred to as the BC Interior) is located to the east of the Coast Mountains and west of the Columbia Mountains and Canadian Rocky Mountains (Figure 1). This region has an arid climate with warm, dry summers and cool, dry winters due to the rain shadow of the Coast Mountains [36]. However, the region experiences

substantial northwest-to-southeast temperature and precipitation gradients, with temperatures increasing (from mean annual temperature of 4.5 °C in Williams Lake to mean annual temperature of 8.1 °C in Kelowna) and precipitation decreasing (from mean annual precipitation of 450 mm/year in Williams Lake to mean annual precipitation of 387 mm/year in Kelowna) towards the southeast of the BC Interior [37]. The hydrological regime is snowmelt-dominated, with peak streamflow occurring in the spring/summer and low flows occurring in late summer through to late winter [38,39]. Fall rains cause a slight increase in streamflow. Monthly rainfall in the BC Interior is the highest in the summer, but due to high temperatures and high evapotranspiration rates [40], groundwater recharge is lowest during the summer months [38,39,41].

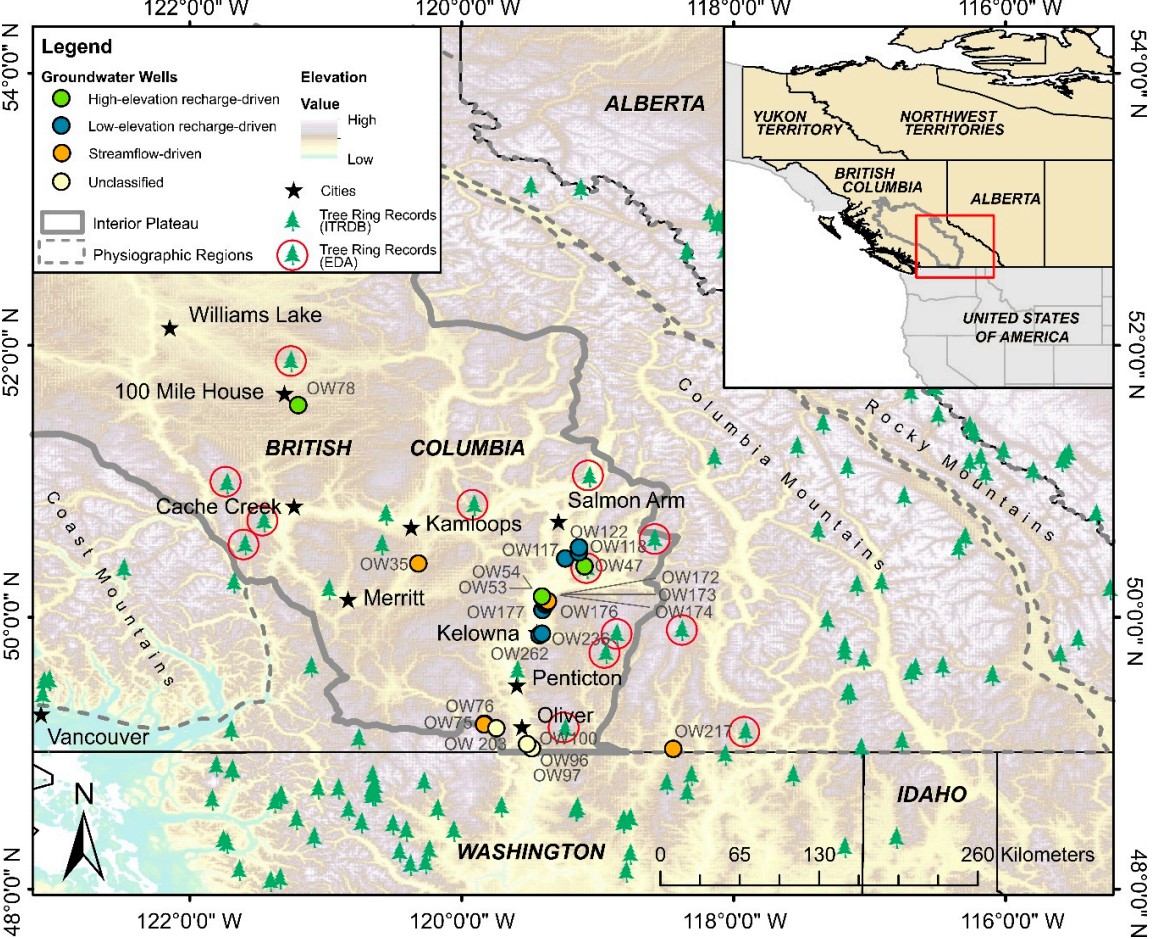

**Figure 1.** Location map of observation wells within the southern Interior Plateau Region of British Columbia and tree ring records available from the International Tree Ring Database (ITRDB). Tree ring records used for the Explorative Data Analysis (EDA) are circled in red. The inset map shows the location of the study region within western Canada and the northwestern United States. Observation wells are colour-coded by aquifer–stream system type and classified using hysteresis plots as discussed in Section 2.1. Source for basemap digital elevation model (DEM): HydroSHEDS Database, available from http://hydrosheds.cr.usgs.gov.

Common tree species in the BC Interior include Engelmann Spruce, Douglas-Fir, Ponderosa Pine, Subalpine Fir, Subalpine Spruce, Subalpine Larch, and Whitebark Pine. These tree species can be found at a range of elevations, corresponding to different biogeoclimatic zones (BGCZs) [42]. The Ponderosa Pine zone is commonly found in the valley bottom (starting at an elevation of about 350 masl (metres above sea level)), Interior Douglas fir at low mountainous elevations, Montane Spruce at mid-elevations, and Engelmann Spruce–Subalpine Fir at the highest mountain elevations (up to

2100 masl). Sub-Boreal Pine–Spruce zones also occur at high elevations, but only on the Interior Fraser and Nechako Plateaus. In these regions, summers are generally warm and dry, with cool, dry winters. All these BGCZs experience snow in the winter, with the higher elevations receiving the most snowpack. The Interior Cedar–Hemlock and Sub-Boreal Spruce zones are also found in the BC Interior, and these zones have moisture available throughout the year and warm, moist growing seasons. Both the Interior Cedar–Hemlock and Sub-Boreal Spruce zones experience winter precipitation in the form of rain and snow, which melts at the beginning of the growing season. In the Sub-Boreal Spruce Zone, summer precipitation often comes in the form of convective storms [43]. The growth of trees found in these BGCZs may be limited by moisture or temperature depending on the environment. In mountain sites, snowpack is a particularly important variable which may limit or enhance tree growth. At lower elevations, snowmelt may be an important source of water to trees during the growing season, and therefore these trees are considered moisture sensitive. At higher elevation sites, a deep, persistent snowpack suppresses tree growth; these trees are sensitive to temperature, as the temperature over the winter and spring determines the amount of precipitation that falls as snow, and how long the snowpack persists into the spring. Both relationships between tree growth and snowpack have been used to reconstruct snowpack and snow water equivalent from tree rings e.g., [11,44–48].

## 2. Materials and Methods

### 2.1. Explorative Data Analysis

#### 2.1.1. Classifying Wells Based on Dominant Recharge Mechanism

Hysteresis plots showing groundwater level versus stream discharge provide a useful means of classifying observation wells based on their dominant recharge mechanism [30]. These plots were generated for all sites in the study region where a stream gauge was located in proximity to a BC observation well (ten observation wells). Groundwater level data were obtained from the BC Observation Well Network [33] and streamflow from Environment and Climate Change Canada (ECCC) hydrometric stations [49]. Hysteresis plots were created in R [50] using a custom script (Paul Whitfield, ECCC, personal communication). The plots were created using daily data for a 3-year period—a period of sufficient length to identify the direction of hysteresis. Data from 2013 to 2015 were used where possible; however, where there were gaps in either the streamflow or groundwater level records for these years, different years were chosen (see Supplementary Information, Figures S1–S3).

The hysteresis plots were used to distinguish between streamflow-driven and recharge-driven aquifer–streams. A negative (or counter-clockwise) hysteresis loop (Figure 2a) represents the streamflow driving groundwater levels; an increase in streamflow is followed by an increase in groundwater level, and a decrease in streamflow is followed by a decrease in groundwater level (streamflow-driven). A positive (or clockwise) loop shows the opposite relationship, where an increase in groundwater level is followed by an increase in streamflow (recharge-driven) (Figure 2b).

Streamflow and groundwater level hydrographs for streamflow-driven systems peak in late spring/early summer due to snowmelt. In contrast, while stream hydrographs in recharge-driven systems also peak in late spring/early summer, the lowest groundwater levels occur in the late summer/ early fall. Recharge-driven systems can exist in both high-elevation and low-elevation areas, and in the case of high-elevation recharge-driven systems, groundwater levels can closely mimic a nearby streamflow hydrograph (and appear as if it is a streamflow-driven system), even though the dominant recharge mechanism is diffuse recharge by snowmelt [30].

Where no nearby streams were present, or the hysteresis plot produced no defined loop structure, wells were classified into streamflow-driven or recharge-driven systems based on the shape of the groundwater hydrograph and the location of the well. Wells with the highest groundwater levels in the summer were classified as streamflow-driven, while wells with the lowest groundwater levels in the late summer/early fall were classified as recharge-driven. Wells that appeared to be streamflow-driven, but are located in high-elevation bedrock aquifers, were classified as high-elevation recharge-driven. Of the

22 available long-term observation wells in the BC Interior, seven were classified as streamflow-driven, four as high-elevation recharge-driven, and seven as low-elevation recharge-driven (Supplementary Information, Figures S4–S7 and Table S1). Three wells (OW96, OW97, OW100) were not classified, as their hydrographs indicated that the dominant recharge mechanism was likely irrigation and therefore did not fit into any of the recharge categories, and one well (OW203) was not classified because the dominant recharge mechanism was not clear from the hysteresis plot.

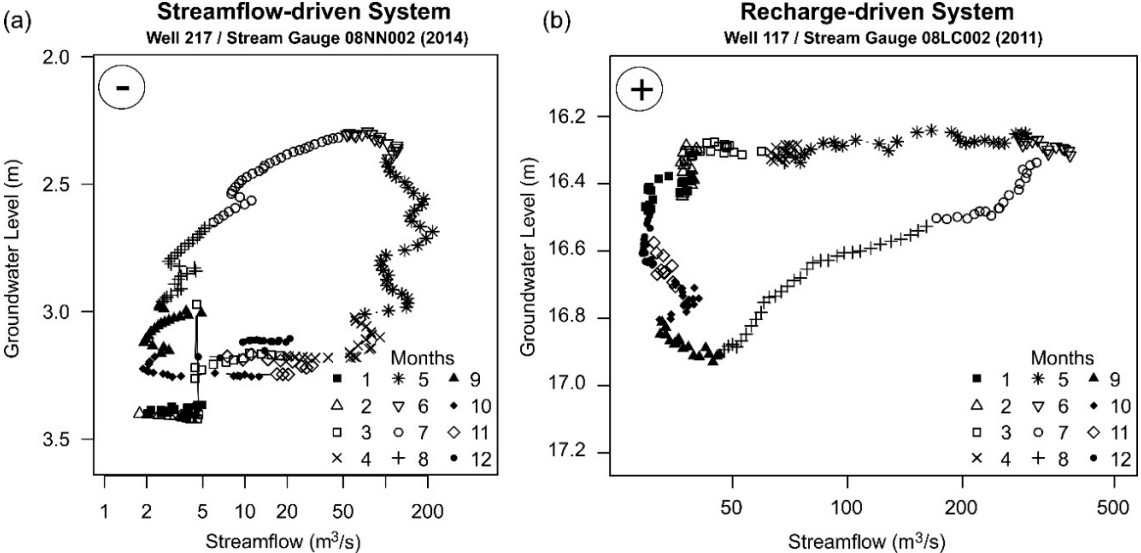

**Figure 2.** Representative hysteresis plots for (**a**) a streamflow-driven system characterized by a negative or counter-clockwise loop, and (**b**) a recharge-driven system characterized by a positive or clockwise hysteresis loop.

### 2.1.2. Tree–Groundwater Relationships

Exploring the relationship between tree growth and groundwater levels requires annual measurements of raw ring width and monthly groundwater level records. Correlations between the groundwater levels and tree ring chronologies were analyzed in R [50]. Measurements of tree ring widths for 15 tree stands in the BC Interior (see Figure 1 for the locations of tree stands used) were downloaded from the ITRDB [34]. Tree ring records uploaded by authors to the ITRDB are assumed to have been checked by the authors to ensure correct age-dating [51]; however, the obtained records of tree ring widths were cross-dated in the program COFECHA [52] to verify the published dates. Each record consists of multiple samples of tree ring widths located within one stand of a single tree species. Species include Douglas-fir (1), Engelmann Spruce (12), and Subalpine Fir (2) (See Supplementary Information, Table S2). These stands range in elevation from 810 to 3800 masl across the BC Interior (Figure 1).

Tree ring chronologies for these sites were developed in R using the package "dplR" [53]. Raw ring measurements for individual trees were first detrended to account for the age-related growth trend using a 2/3 cubic smoothing spline with a 50% frequency cutoff [54,55]. For the few individual series where the cubic spline resulted in a negative growth curve, a negative exponential curve was used. The cubic smoothing spline removes 50% of the variance in the time series at a frequency equal to two-thirds (67%) of the length of each individual series. This process is intended to remove very low frequency variations, while leaving the high-frequency variability due to climate intact. The raw ring widths were then standardized by dividing each measurement by the value of the detrending curve (ratio method). The resulting values are referred to as ring width indices and are positive values with a mean of 1. Ring width indices from each tree at a site were then averaged using Tukey's biweight robust mean [56,57] to create one chronology for each tree ring site. This is referred to as the standard version of chronology and still includes any autoregression found in the timeseries. In dplR, a residual

chronology can also be computed to remove autocorrelation from the time series, but because the groundwater levels were found to be autocorrelated, we chose to use the standard chronologies for this portion of the analysis (Supplementary Information, Figures S8–S11).

Monthly depth to groundwater level data (hereafter referred to as groundwater level records) were collected from the BC Provincial Groundwater Observation Well Network [33]. Observation wells were included if they had at least 30 years of observed data and showed no influence from human activities (such as declining groundwater levels or erratic seasonal trends). Seventeen observation wells met these criteria; however, four additional observation wells that showed a historical declining trend were also analyzed, as they were potentially representative of a low elevation recharge-dominated regime (Supplementary Information, Table S1). The declining historical trend in these four wells was removed using cubic detrending in the U.S. Geological Survey software HydroCLIMate [58].

The entire period of overlap between the groundwater level data and the ring width indices was used to determine tree growth-groundwater relationships. Each observation well was paired with up to five of the tree ring sites from the southern BC Interior for the correlations (distances on the order of 100 km or less). The period of overlap varied for each pair, ranging from 17 to 32 years. For this analysis, monthly groundwater levels were correlated with annual ring width indices using the R package "treeclim" [59] to identify the months with the strongest correlations to tree ring growth (Supplementary Information, Figures S12–S23). These correlations were used to determine the appropriate months to target for the groundwater level reconstructions. As tree growth occurs seasonally (predominantly during the spring/summer months), targeting a specific set of months for the reconstruction helps to identify the time of year where groundwater level records are most strongly correlated to the annual ring widths, which are related to the climate variables that limit tree growth (see Section 3.1). These months were chosen by identifying months with significant positive correlations ($p < 0.05$) between depth to groundwater level and tree ring widths in each of the observation well-tree ring record pairs (see Section 3.1). This correlation analysis was done individually for the 22 wells collected from the southern BC Interior, as well as for all wells combined (i.e., no distinction according to recharge mechanism). The months from the previous year spring (April) to current year fall (October) were analyzed for the individual observation well-tree ring record pairs, and from the previous April to the current November for the average of all wells' time series. As the all-wells average was composed of wells in various aquifer–stream systems, this longer period was used to cover both the periods of highest and lowest groundwater levels. The same analysis was also conducted using correlations withing moving time windows to determine the time-stability of the relationships between the groundwater levels and the tree ring chronologies (Supplementary Information, Figures S24–S29). Twenty-year moving windows were used if the data allowed; in some cases where the period of overlap for an individual observation well-tree ring record pair was short, a shorter moving window had to be used, because the treeclim code will not run when the number of years of overlap is shorter than the number of variables being tested [59]. Due to the short period of overlap of some observation well-tree ring record pairs, the moving correlation plots were not possible for all pairs (moving time correlations were conducted for 33 pairs).

## 2.2. Selecting the Appropriate Tree Ring Network

Of the 15 tree ring stands available in the BC Interior, only one (Cana 229) has a long enough period of overlap (>30 years) with the groundwater level records that it could be used as part of a reconstruction model. However, tree growth is thought to respond to regional-scale climate variations (on the order of hundreds of kilometres), allowing for the use of a wider network of tree ring records for use in reconstructions [31]. Two different approaches were used in this study to select which tree ring records could be used as predictors in the model to reconstruct BC Interior groundwater levels.

The first method involved creating a climate footprint, or a spatially defined area with a common climate signal [13,20]. All the tree ring records within this climate footprint will not necessarily be correlated to groundwater levels due to variations in species, elevation, and location and therefore

cannot be considered as predictors of groundwater level response. Therefore, these tree ring records are screened during the modeling process to look for a significant correlation between groundwater levels and ring width for use in the model (see Section 3.2). For this study, the Koninklijk Nederlands Meteorologisch Instituut (KNMI) Climate Explorer was used to create a climate footprint, which is a spatial correlation field between average groundwater levels in the BC Interior and the Consejo Superior de Investigaciones Científicas (CSIC) global 0.5° gridded 12-month Standardized Precipitation Evaporation Index (SPEI) dataset (Figure 3) [60]. While SPEI is a drought signal, it defines an area of similar precipitation and evaporation, thus explicitly taking temperature into account. The climate footprint delineates a region where climate influences are expected to be similar. While local areas within this climate footprint may have varying climate responses, the climate footprint defines a region of similar regional response. This method is suitable for the principal component regression method used for the groundwater level reconstructions (see Section 2.3), as the principal components regression analysis helps to extract common regional climate signals from the available tree ring chronologies. This correlation analysis identifies a region where the groundwater level records are significantly correlated ($p < 0.10$) to the gridded SPEI. This significance is indicative of the average significance of all maps generated by the KNMI Climate Explorer, although the significance of the correlation field (the "field significance") is also estimated by the software based on the size and location of the significantly correlated grid cells [61]. The field significance for the correlations between groundwater levels and 12-month SPEI was $p < 0.01$. Grid cells were included in the climate footprint if the correlation coefficient was greater than 0.3. The correlation was performed over the months identified in Section 2.2 as the months with the strongest correlations to tree ring widths for each of the three categories of recharge mechanism and for an average of all wells, and then superimposed to create one climate footprint. The SPEI was chosen to define the climate footprint because it can characterize drought at multiple time scales (from 3 to 48 months). The 12-month lag for SPEI was chosen because groundwater levels have been shown to be related to the SPEI at longer time scales (6–24 months) [61]. A total of 118 tree ring records were found within this SPEI climate footprint covering the period 1900–2004.

The second method of selecting tree ring records was by ecological region (Supplementary Information, Figure S30). North American Ecological Regions (ecoregions) [35] are based on a hierarchical classification system which incorporates stable ecosystem characteristics such as vegetation, landforms and soil type, climate and biota. Therefore, ecoregions define regions where similar growing conditions are expected to occur. Ecoregions provide an alternative to the climate footprint (defined by a drought index), which may be better suited for mountain regions. Level 2 ecoregions are meant to be used for regional scale analyses. Three Level 2 ecoregions found in southern BC and extending southwards into the United States were used to select potential tree ring records: Mountain Ecoregions, Plateau Ecoregions, and Coast Mountain Ecoregions. These ecoregions contained 144, 27, and 21 tree ring records, respectively, covering the period from 1900 to 2004. A smaller subset of each ecoregion was also considered (see Section 2.3).

Tree ring widths for each record within the designated climate footprint (Figure 3) and the three ecoregions (Coast Mountain, Plateau, and Mountain) were downloaded from the ITRDB [32]. All available species within each region were considered (Supplementary Information, Table S4). To develop site chronologies, raw ring width measurements for individual trees were detrended, standardized, and averaged in ARSTAN [56] using the methods described in Section 2.1.2. The advantage of using ARSTAN to create the final chronologies used for the reconstructions is that ARSTAN also uses autoregressive modeling to compute an ARSTAN chronology (in addition to the standard and residual chronologies), which adds persistence back into the residual chronology by adding back the pooled autoregression model (i.e., the autoregression that is common among all the individual trees) to the residual chronology. The ARSTAN chronology was chosen for the final chronologies used for modeling because it is meant to strengthen the climate signal in the chronologies, and the relationship between the tree ring records and hydroclimate was of interest in this study [13,20,56]. Additionally,

as the groundwater levels were found to be autocorrelated, adding a portion of the autoregression back into the tree ring chronologies would be useful for reconstructing autocorrelated time series. The quality of each chronology was assessed using the expressed population signal (EPS), a measure of chronology uncertainty due to a decreasing number of cores further back in time [62,63]. A chronology is typically truncated when the EPS falls below a set value to remove portions of the chronology where the chronology signal is no longer able to portray a theoretical chronology due to a decreasing number of tree cores [64]. Chronologies used in this study were truncated when the EPS fell below 0.8 (with an EPS of 1.0 being the maximum value).

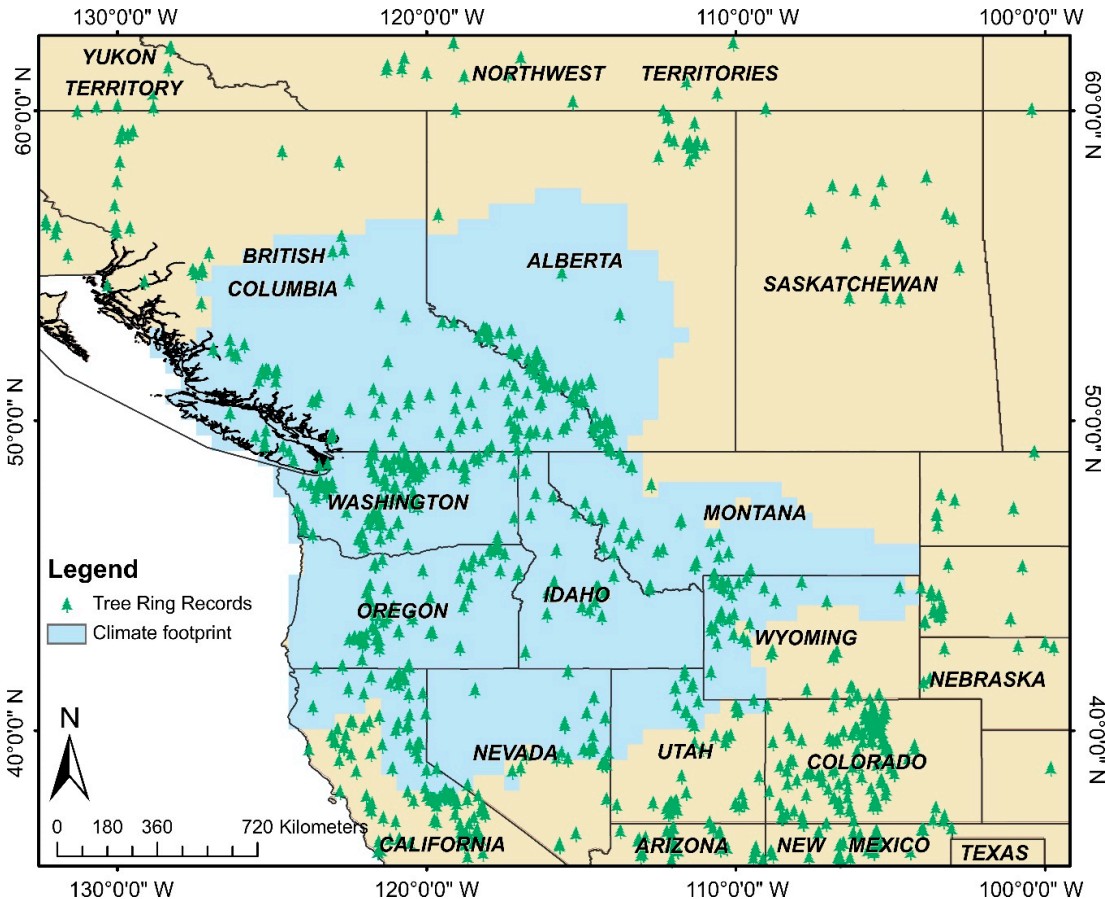

**Figure 3.** Climate footprint for the BC Interior showing the spatial extent of significant correlation ($p < 0.10$) between average depth to groundwater levels and gridded 12-month Standardized Precipitation Evaporation Index (SPEI) (Koninklijk Nederlands Meteorologisch Instituut (KNMI) Climate Explorer).

*2.3. Groundwater Level Reconstructions*

To assess the effectiveness of the different methods for reconstructing groundwater levels, 28 reconstructions of average groundwater levels were created for the time period of 1900 to 2004, using the seven regional subsets of tree ring chronologies (described below) and four different calibration times series derived from the observation well data.

The four calibration time series were assembled based on recharge mechanism and included: (1) an average of all available wells (22 wells) in the southern BC Interior, (2) an average of the wells in streamflow-dominated systems (7 wells), (3) an average of all wells in high-elevation recharge-dominated systems (4 wells), and (4) an average of all the wells in low-elevation recharge-dominated systems (7 wells). All groundwater level records were standardized prior to calculating the average for each category by subtracting the mean of each record from each monthly measurement and then dividing by the standard deviation.

Seven scenarios were tested for each category of recharge mechanism using different regional subsets of the tree ring chronologies. This was done to establish which tree ring records were used as the basis for the 28 reconstructions (Table 1). These regional subsets included the four regions described above, (1) the SPEI climate footprint, (2) Mountain Ecoregion, (3) Coast Mountain Ecoregion, and (4) the Plateau Ecoregion), as well as three smaller portions of the ecoregions, specifically, (5) the northern portion of the Mountain Ecoregion, (6) the northern portion of the Plateau Ecoregions (which excludes chronologies south of Oregon, Idaho, and Wyoming), and (7) the southern portion of the Coast Mountain Ecoregion (which excludes chronologies from Alaska).

Reconstruction models were created using the principal component regression program PcReg [65]. PcReg has an advantage over multiple linear regression models, as it extracts the common climate signals from all the site chronologies used and uses these principal components as model predicters, which are representative of the regional changes in climate [66]. This is useful for the interior of B.C., where only one of the local tree ring records was long enough to use for reconstruction modeling, and more regional tree ring chronologies need to be used. The initial groundwater reconstructions were created for the period 1900–2004 (referred to hereafter as the common period), using tree ring widths from all tree ring chronologies which span the entire time period as potential predictors for the model. This time period was chosen to allow for enough overlap between the tree ring records and the groundwater level records (at least 30 years), and because after the year 2004, the number of available tree ring chronologies begins to decrease. Tree chronologies were only selected as predictors that contributed to the reconstruction model if they had a significant ($p < 0.05$) positive correlation to the average groundwater levels record being modeled, with a minimum of 1 tree ring record needed to create a reconstruction model. A statistical procedure known as pre-whitening was used on both the groundwater level records and the tree ring chronologies prior to modeling to remove the effects of autocorrelation from the timeseries. This strengthens the regression procedure, as one of the assumptions for linear regression is that the timeseries are not autocorrelated. The dominant principal components (PCs) were then extracted from this pool of predictors, with PCs being included in the model if the eigenvalue was greater than one [67,68]. The final PCs then underwent stepwise multiple linear regression, and the strongest model was selected using the minimum Aikeke Information Criterion (AIC) [69]. The AIC balances the "goodness of fit" of the model and the complexity of the model and includes a penalty for models with more parameters to guard against overfitting. Finally, the autoregression modelled in the groundwater level time series prior to multiple linear regression modelling was added back into the reconstruction (the reddened reconstruction).

Each model was assessed using a two-thirds split calibration–validation approach, where the earliest two-thirds of the overlap period were used to calibrate the model, and the later one-third of the overlap period was used to validate the model (the forward model). The calibration periods varied between the streamflow-driven (1969–1992), high-elevation recharge-driven (1970–1993), low-elevation recharge-driven (1972–1993), and all-wells (1969–1992) categories, due to the differing periods of record of the individual wells in each category. The calibration and validation periods were then switched (calibration was performed on the later two-thirds, and validation was performed on the first one-third of the overlap period) to assess if the models were stable over time (the reverse model). The reduction of error (RE) and coefficient of efficiency (CE) statistics were used to determine model skill [31]. For a model to be considered time-stable, both statistics needed to be above zero for both calibration periods. The percent variance explained by the reconstruction and Pearson's correlation coefficient (Pearson's r) were also examined as measures to show the strength of each reconstruction model. In multiple linear regression models, the percent variance explained is equal to the $r^2$ value and indicates how well the model explains the variance in the observed groundwater levels. The adjusted percent variance (adjusted $r^2$) is also reported, which is a modified percent variance which accounts for the number of predictors used and the number of samples available for each reconstruction model. Pearson's r is a measure of the strength of the association between the reconstruction and the groundwater level records.

**Table 1.** Model statistics for the 28 model simulations tested using varying recharge mechanisms and varying spatial areas used to select potential tree ring records to be used in the models. All the reconstructions cover the time period 1900–2004. The reduction of error (RE) and coefficient of efficiency (CE) statistics and Pearson's r are shown for the calibration and validation periods (calibration/validation) when calibration was performed on the first two-thirds of the overlap period. Percent variance is shown for the calibration period, with the percent variance values adjusted for the number of degrees of freedom shown in brackets. Models that have positive (>0) RE and CE statistics when calibration was performed on both the early two-thirds and the later two-thirds of the overlap period are shown in bold (i.e., they are time-stable).

| | All Wells (May–Nov) | Streamflow-Driven (July) | High-Elevation Recharge-Driven (June–August) | Low-Elevation Recharge-Driven (August–October) |
|---|---|---|---|---|
| **Climate footprint** | RE: 0.503/0.411<br>CE: 0.503/0.291<br>r: 0.709/0.587<br># tree ring records: 31<br>% variance (adjusted): 50.27 (47.78) | **RE: 0.452/0.247**<br>**CE: 0.452/0.232**<br>**r: 0.673/0.533**<br>**# tree ring records: 30**<br>**% variance (adjusted): 45.29 (39.21)** | RE: 0.375/0.245<br>CE: 0.375/0.109<br>**r: 0.621/0.339**<br>**# tree ring records: 28**<br>**% variance (adjusted): 38.56 (35.49)** | RE: 0.540/0.291<br>CE: 0.540/0.245<br>r: 0.754/0.714<br># tree ring records: 28<br>% variance (adjusted): 56.85 (51.46) |
| **Coast Mountain** | **RE: 0.412/0.496**<br>**CE: 0.412/0.394**<br>**r: 0.633/0.833**<br>**# tree ring records: 6**<br>**% variance (adjusted): 40.07 (37.07)** | RE: 0.236/0.250<br>CE: 0.236/0.235<br>r: 0.489/0.658<br># tree ring records: 6<br>% variance (adjusted): 23.91 (20.10) | RE: 0.220/0.349<br>CE: 0.220/0.231<br>r: 0.470/0.64<br># tree ring records: 6<br>% variance (adjusted): 22.09 (18.19) | RE: 0.489/0.009<br>CE: 0.489/−0.055<br>r: 0.765/0.804<br># tree ring records: 7<br>% variance (adjusted): 58.52 (56.21) |
| **South Coast Mountain *** | **RE: 0.412/0.496**<br>**CE: 0.412/0.394**<br>**s r: 0.633/0.833**<br>**# tree ring records: 6**<br>**% variance (adjusted): 40.07 (37.07)** | RE: 0.236/0.250<br>CE: 0.236/0.235<br>r: 0.489/0.658<br># tree ring records: 6<br>% variance (adjusted): 23.91 (20.10) | RE: 0.220/0.349<br>**CE: 0.220/0.231**<br>**r: 0.470/0.64**<br>**# tree ring records: 6**<br>**% variance (adjusted): 22.09 (18.19)** | RE: 0.489/0.009<br>CE: 0.489/−0.055<br>r: 0.765/0.804<br># tree ring records: 7<br>% variance (adjusted): 58.52 (56.21) |
| **Plateau** | RE: 0.463/−2.24<br>CE: 0.463/−2.895<br>r: 0.685/−0.31<br># tree ring records: 13<br>% variance (adjusted): 46.92 (41.02) | RE: 0.431/−0.972<br>CE: 0.431/−1.012<br>r: 0.657/0.169<br># tree ring records: 10<br>% variance (adjusted): 43.16 (36.84) | RE: 0.032/−1.659<br>CE: 0.032/−2.139<br>r: 0.383/−0.305<br># tree ring records: 10<br>% variance (adjusted): 14.67 (5.19) | RE: 0.727/−3.887<br>CE: 0.727/−4.207<br>r: 0.853/−0.325<br># tree ring records: 13<br>% variance (adjusted): 72.76 (69.35) |
| **North Plateau** | No model possible | RE: 0.254/−0.471<br>CE: 0.254/−0.501<br>r: 0.510/0.135<br># tree ring records: 1<br>% variance (adjusted): 26.01 (22.31) | No model possible | No model possible |

**Table 1.** *Cont.*

|  | All Wells (May–Nov) | Streamflow-Driven (July) | High-Elevation Recharge-Driven (June–August) | Low-Elevation Recharge-Driven (August–October) |
|---|---|---|---|---|
| **Mountain** | RE: 0.684/−0.156<br>CE: 0.684/−0.390<br>r: 0.829/−0.031<br># tree ring records: 40<br>% variance (adjusted): 68.72 (67.16) | RE: 0.463/0.081<br>CE: 0.463/0.062<br>r: 0.681/0.300<br># tree ring records: 34<br>% variance (adjusted): 46.38 (43.70) | RE: 0.485/−0.143<br>CE: 0.485/−0.35<br>r: 0.702/−0.074<br># tree ring records: 35<br>% variance (adjusted): 49.28 (46.74) | RE: 0.912/−0.388<br>CE: 0.912/−0.479<br>r: 0.956/0.120<br># tree ring records: 38<br>% variance (adjusted): 91.39 (90.31) |
| **North Mountain** | RE: 0.520/0.316<br>CE: 0.520/0.178<br>r: 0.721/0.444<br># tree ring records: 25<br>% variance (adjusted): 51.98 (49.58) | RE: 0.411/0.109<br>CE: 0.411/0.091<br>r: 0.641/0.343<br># tree ring records: 23<br>% variance (adjusted): 41.09 (34.54) | RE: 0.422/0.102<br>CE: 0.422/−0.06<br>r: 0.660/0.203<br># tree ring records: 23<br>% variance (adjusted): 43.56 (40.74) | RE: 0.647/0.319<br>CE: 0.6470.274<br>r: 0.822/0.686<br># tree ring records: 24<br>% variance (adjusted): 67.57 (65.77) |

* Note: the models created for the South Coast Mountain Ecoregion are identical to the models created using the entire Coast Mountain Ecoregions to select tree ring records, despite excluding chronologies from Alaska.

Extended reconstructions were created for each model that was found to be time-stable. Model nests were used to extend these reconstructions further back in time, with 50-year reconstruction time periods (nests) being added on to the original common period model. For example, the first model nest back in time covered the period from 1850 to 2004, and the first 50 years from that model were added on to the beginning of the common period reconstruction, which covered the period from 1900 to 2004. This nesting technique compensates for the fact that more chronologies drop out further back into time and allows for the maximum number of tree ring records to be used for each time period of the model [13,70]. All reconstruction nests were standardized to the mean and variance of the observed groundwater level records by subtracting the reconstruction mean from each value and dividing by the standard deviation of the reconstruction, and then adding the observed record mean and multiplying by the observed standard deviation.

## 3. Results

### 3.1. Tree Ring-Groundwater Relationships

The correlations between groundwater levels and tree ring width showed that the most common relationship between the two variables was a positive correlation during the current year summer and fall, with significant correlation coefficients ranging from 0.19 to 0.72 (Supplementary Information, Table S3). Of the 42 groundwater well-tree ring record pairs that show significant monthly correlations, only 5 pairs produced negative correlations, and they all occurred when Cana 150 and Cana 152 (two Engelmann Spruce chronologies found at an elevation of 1700 m) were used in the pairings, with these negative correlations occurring earlier in the year than the majority of the other pairings (during the winter/spring months—see Supplementary Information, Table S3). Because these negative relationships appeared to be anomalous and potentially only related to these two tree ring chronologies, only positive correlations were considered for the reconstruction models. Additionally, while some pairs did show significant correlations during the previous year, no recurring patterns in the previous year relationships were identified; therefore, only the current year relationships are considered.

The most consistent pattern among the correlations was a positive correlation between depth to groundwater levels and current-year tree ring width, which corresponds to the timing of peak groundwater levels in the streamflow-driven (and high-elevation recharge-driven) systems, and the timing of lowest groundwater levels for the low-elevation recharge-driven systems (Figure 4, Supplementary Information, Table S3). These time periods correspond to the summer months for high-elevation recharge-driven and streamflow-driven systems, and to the late summer/early fall for the low-elevation recharge-driven systems. The analysis of these correlations over moving windows of time suggest that, overall, this relationship remains relatively stable over time, with the sign of the correlations in the late summer/early fall months remaining positive throughout all of the 20-year windows for 30 out of the 33 pairs tested using the moving window correlation analysis (Supplementary Information, Figures S24–S29). The relationship is therefore considered sufficiently stable through time to use for reconstructions of groundwater levels.

As the peak groundwater levels in the streamflow-driven and high-elevation recharge-driven systems are related to snowmelt, these results suggest that it is the previous winter's snowpack which connects the tree ring widths to the groundwater levels; when there is a larger snowpack, tree growth in mountainous sites is limited because the snow persists longer, and the length of the growing season is reduced (so smaller tree rings are formed). This corresponds to increased snowmelt and higher summer groundwater levels (or smaller depth to groundwater levels), leading to the positive correlation between the two variables. For the low-elevation recharge-driven systems, it appears that the connection to snowpack may be less important, as the tree ring widths are correlated to the timing of lowest groundwater levels. This suggests that in years with a shorter growing season, early fall groundwater levels are higher (the depth to groundwater levels are smaller). However, in the low-elevation recharge-driven systems, higher groundwater levels in the early fall are expected to be

driven more by the amount of precipitation over the summer and early fall than by snowpack. It should be noted that while trees in the southern interior of British Columbia begin to go dormant in the fall months, the lag between the growing season climate and the response of the groundwater system means that the annual growth rings can still be related to the groundwater levels during the fall months. Therefore, the summer (for high-elevation recharge and streamflow-driven) and late summer/early fall (low-elevation recharge driven) months were chosen as the targets for the groundwater level reconstructions where the wells are divided by recharge mechanism.

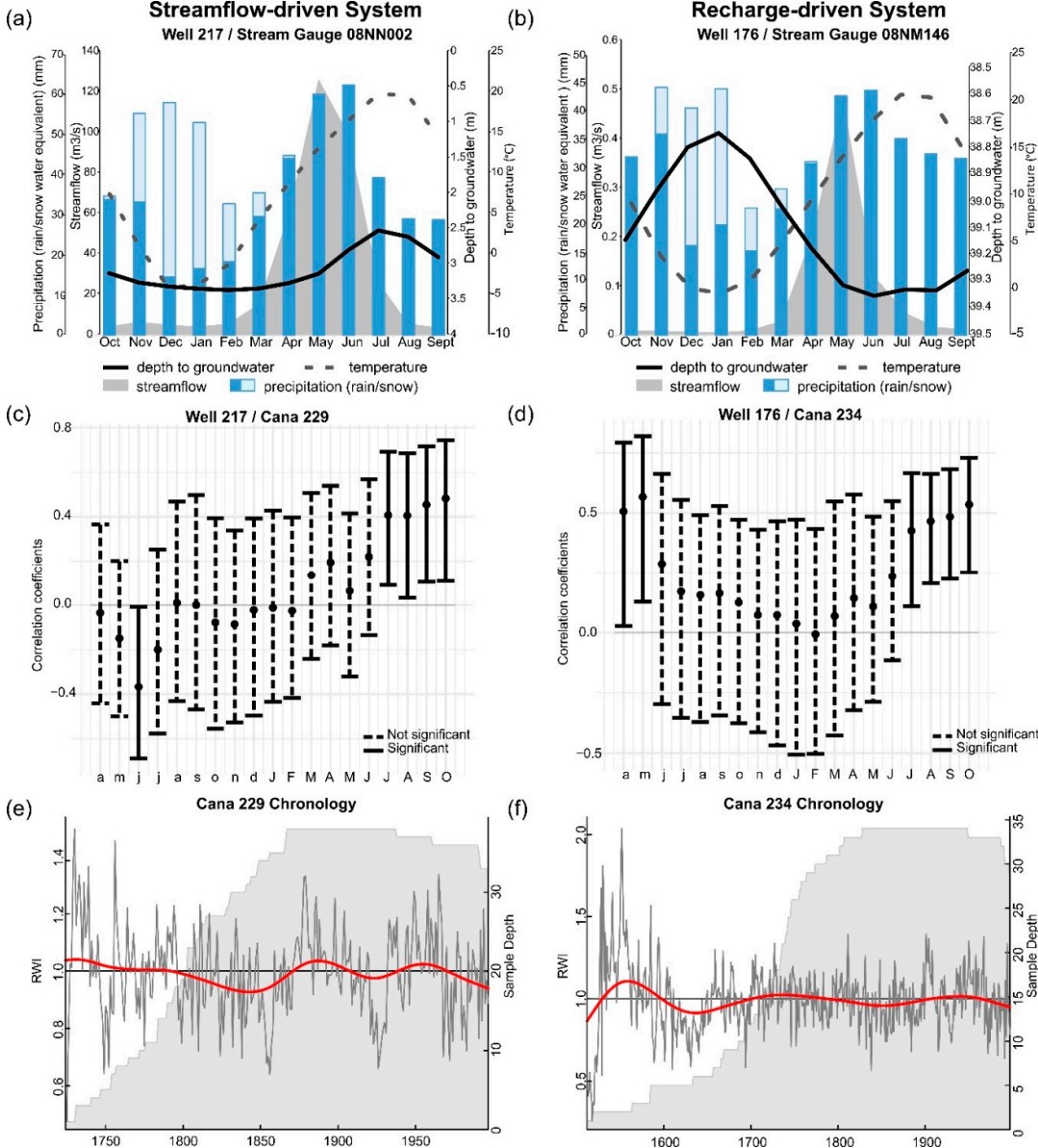

**Figure 4.** Example groundwater (black solid line) and streamflow (shaded grey area) hydrographs for (**a**) a streamflow-driven system and b) a recharge-driven system. Temperature (grey dashed line) and precipitation (blue/light blue bars distinguishing rain from snow) are also shown in a) and (**b**). Corresponding tree ring width/depth to groundwater level correlation graphs are shown for (**c**) a streamflow-driven system and (**d**) a recharge-driven system. The standard chronologies produced using dplR are shown for the two sample chronologies which were located near these observations wells: (**e**) Cana 229 and (**f**) Cana 234, with the grey solid line indicating annual ring width indices, grey shading indicating the sample depth through time, and the solid red line showing a 20-year smoothing spline.

When the average of all the wells was correlated to the tree ring records, significant correlations were consistently found during both the summer and early fall, suggesting influence from both the streamflow-driven and recharge-driven regimes (Supplementary Information, Table S3). Of the ten tree ring records that showed significant correlations to the average groundwater levels, five also showed significant correlations starting in the winter. However, based on the relationships identified when the wells were assessed individually, these significant correlations during the winter months may not be representative of the overall relationship between the tree ring records and the average groundwater levels. Therefore, the period from the early summer to early fall was chosen as the target for reconstruction of the all-wells average groundwater levels.

These relationships were also tested in the pcReg (see Section 2.3) to ensure that this program also identified these months as having significant correlations between the ARSTAN tree ring chronologies and the average groundwater level time series used to calibrate the models. Based on the results of the exploratory data analysis, the final months chosen for the reconstructions were: July for the streamflow-driven systems, June-August for the high-elevation recharge-driven systems, August–October for the low-elevation recharge-driven systems, and May–November for the regional average of all the available observation wells.

### 3.2. Model Comparisons

Twenty-eight groundwater level reconstruction models were created to show the effects of dividing groundwater level wells based on the dominant recharge mechanism and of using various spatial extents to help select tree ring records to use in the reconstruction models (Table 1). Of the models tested, only six have positive RE and CE statistics for both of the calibration periods (indicating they are time-stable): the two streamflow-driven and high-elevation recharge-driven models using the climate footprint, and the four all-wells and high-elevation recharge-driven models using the Coast and South Coast Ecoregions. However, the models created using the South Coast Ecoregions are identical to those created using the Coast Ecoregions, which suggests that even when the entire Coast ecoregion area is used, only tree ring records in the southern portion have significant positive correlations with calibration time series and therefore are selected for use in the model. All models using the Plateau Ecoregion have negative RE and CE statistics, suggesting that models produced using tree ring chronologies from this region are not skillful at reconstructing groundwater levels. Additionally, when only the northern region of the Plateau Ecoregion is used, so few tree ring records are significantly correlated to the groundwater level calibration time series that a reconstruction model cannot not be created. One exception is a reconstruction model based on the streamflow-driven calibration time series, which was created using only one predictor (no principal components analysis is performed when one predictor is used). Using the Mountain and North Mountain Ecoregions produced reconstruction models with varying levels of skill (with more of the North Mountain Ecoregion models showing positive RE and CE statistics); however, none of these models are time-stable and all lack positive RE and CE statistics for both calibration periods. These results suggest that the climate footprint approach for selecting tree ring records is the most consistent for producing skillful and time-stable reconstructions when the wells are separated into calibration time series based on recharge category. However, when all wells are considered, the Coast Mountain Ecoregions produce models with stronger RE and CE statistics. None of low-elevation recharge-driven models are time-stable.

### 3.3. Tree Ring Records Used for Different Recharge Mechanisms

As the climate footprint was the only scenario to produce time-stable models for two of the three categories of recharge mechanism, these models were analyzed in further detail. Specifically, the spatial distribution of tree ring records selected for the three reconstruction models was examined, to see if their locations differed depending on the type of recharge mechanism used to define the calibration time series. Using the climate footprint, these three models selected 30 (streamflow-driven model), and 28 (low-elevation and high-elevation recharge-driven model) tree ring chronologies that

are significantly correlated to each groundwater level calibration time series for the 1900–2004 period (Figure 5). While there is considerable overlap between the tree ring records used for all three models, some notable differences also exist. The high-elevation recharge-driven model does not include one chronology from Washington (WA135), and instead includes an additional chronology from Wyoming that is not included in the other two models (WY046). The streamflow-driven model includes two additional chronologies from Wyoming (WY041) and Utah (UT535) that are not used in either of the other two models. Finally, the low-elevation recharge-driven model includes one additional chronology from Montana (MT120) instead of one from Oregon (OR091) (Supplementary Information, Tables S5–S7). Of the combined tree ring records used for the three groundwater level reconstruction models, six are found in the Coast Mountain ecoregions and 26 are found in the Mountain ecoregions. None are found in the Plateau ecoregions.

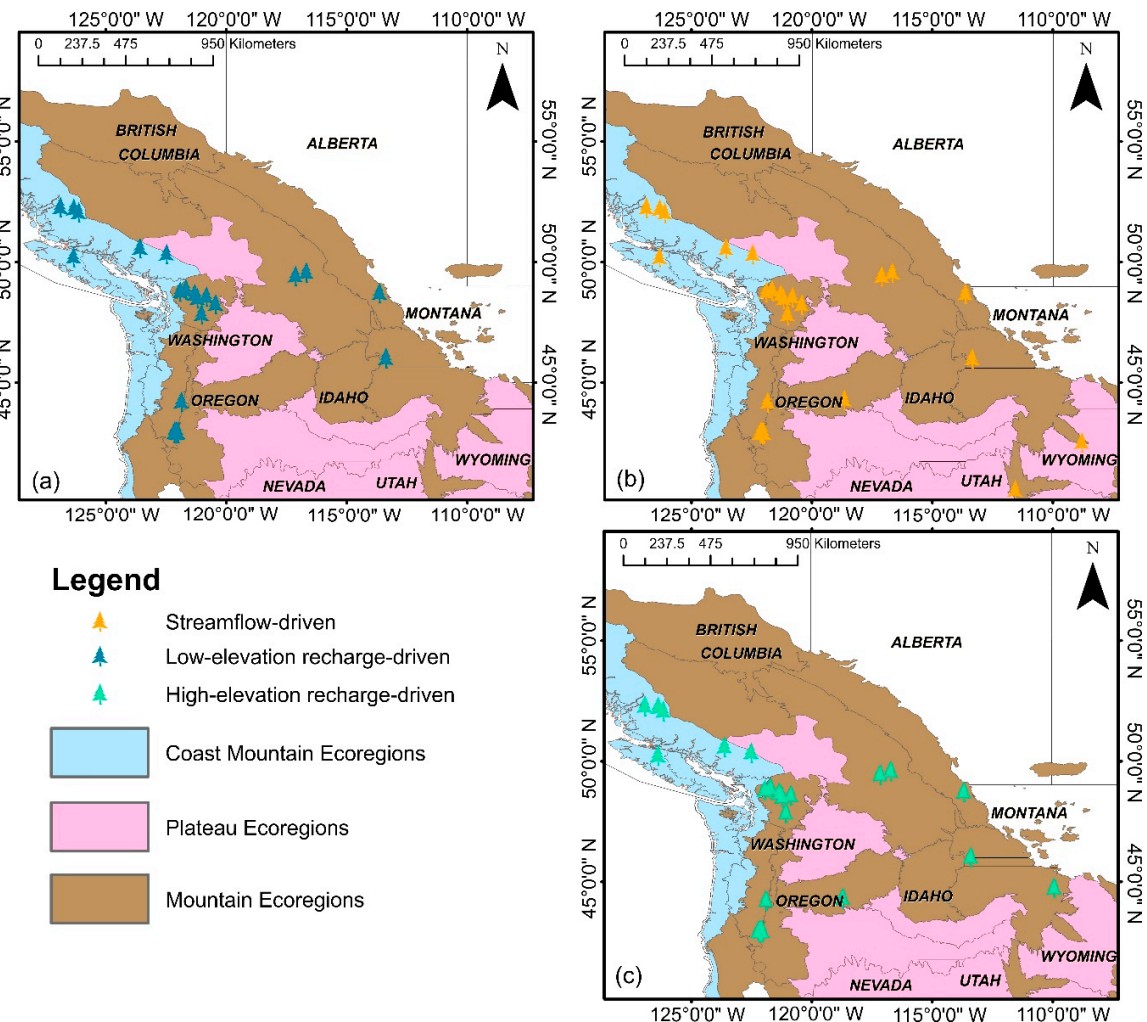

**Figure 5.** Tree ring records used in the (**a**) low-elevation recharge-driven, (**b**) streamflow-driven, and (**c**) high-elevation recharge-driven groundwater level models, using the climate footprint to select potential tree ring records.

### 3.4. Final Groundwater Level Reconstructions

Extended reconstructions were created using the four models that appeared to be time stable (models shown in bold in Table 1): the streamflow-driven and high-elevation recharge-driven models using the climate footprint to select tree rings, and the all-wells and high-elevation recharge-driven models using the Coast Mountain Ecoregion to select tree rings (Figure 6). These models extend the groundwater levels back to the years 1700, 1500, 1750, and 1500, respectively. The reconstructions

were ended when the number of tree ring records was insufficient to create a model (in the case of the high-elevation recharge-driven model using the Coast Mountain Ecoregions) or when two model nests in a row showed negative RE and/or CE values. The high-elevation recharge-driven model created using the climate footprint has a period from 1800 to 1850 where the CE falls below zero, but there are subsequent later periods which still have positive RE and CE statistics, so these later periods were retained within the model (see Supplementary Information, Figure S31).

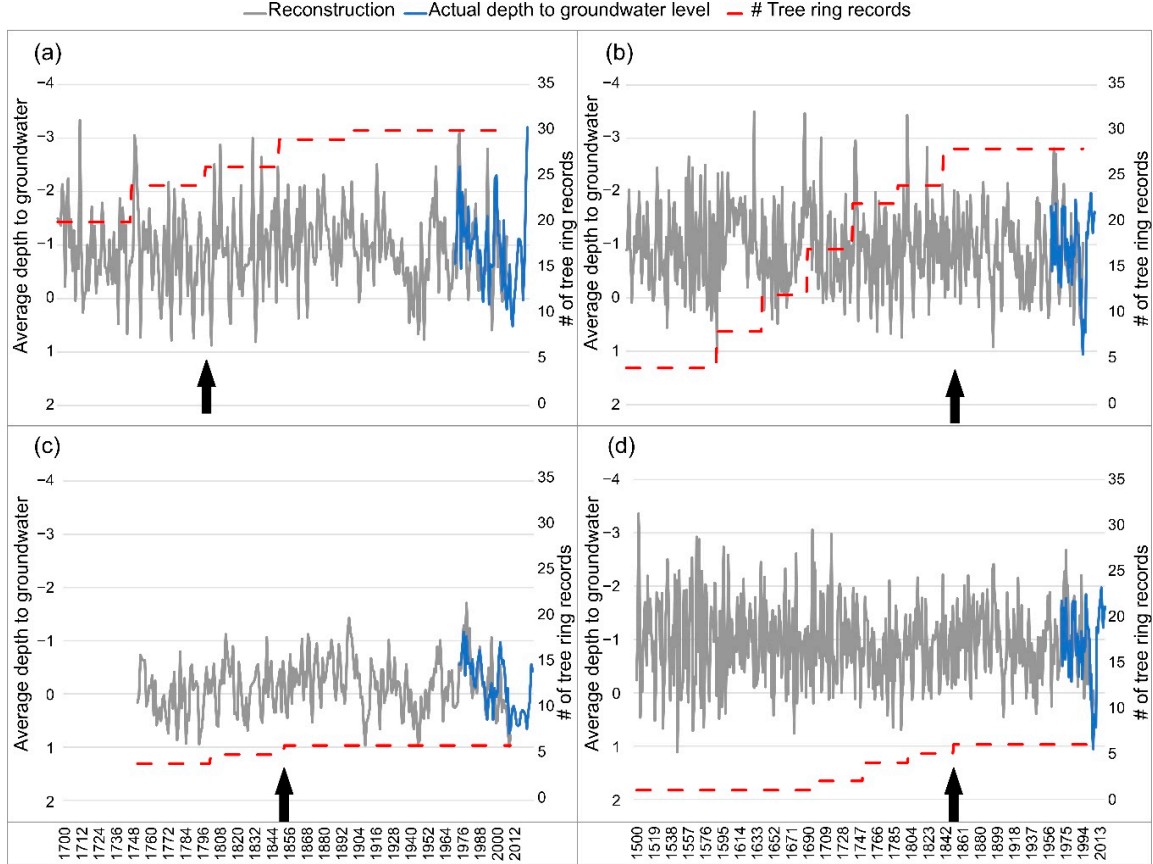

**Figure 6.** Final groundwater level reconstructions and actual groundwater levels (GWL, shown in dark blue) for the average of wells in the (**a**) streamflow-driven and (**b**) high-elevation recharge-driven models using the climate footprint, and the (**c**) all-wells and (**d**) high-elevation recharge-driven models created using the Coast Mountain Ecoregions. The number of tree ring records used in each model throughout time is shown with the dashed red line. The black arrow indicates the time at which the reconstruction models might be truncated due to increasing uncertainty moving farther back into time (see Section 4).

## 4. Discussion

One goal of this study was to determine if using an average of all the available groundwater wells for groundwater level reconstructions was appropriate, given the varying aquifer–stream systems that can be found within mountainous regions. The model comparisons (Table 1) show that dividing wells into categories based on the dominant recharge mechanism only strengthened the reconstruction models when the climate footprint was used to define the region in which to select potential tree ring records. While the all-wells model has the highest RE (0.411) and CE (0.291) during the validation period, these values reflect only the forward modeling results; the reverse model is not robust (i.e., when the calibration and validation periods are reversed the RE and CE values are negative), indicating that the relationship between groundwater levels and the tree ring widths is not time-stable. This could be because the low-elevation recharge-driven model is also not time-stable, and therefore these wells

have a negative effect on the model when all wells are combined. The weaker relationship between low-elevation recharge-driven groundwater levels and tree rings is discussed further below.

Considering the results by ecoregion, only the Coast and South Coast Ecoregions each had two models that are time-stable; the models using all-wells have stronger statistics compared to the high-elevation recharge-driven models. For all other ecoregions, no models meet the time-stable criterion either for all-wells or wells categorized by recharge mechanism. Overall, the modeling results suggest the relationship between groundwater levels and tree ring widths may be stronger when groundwater wells are categorized, but only in the case when the larger climate footprint is used to reconstruct streamflow-driven and high-elevation recharge-driven groundwater levels. Reasons for this relationship are discussed below.

Analyzing the effects of different spatial areas to select tree ring records identified two outcomes: first, that using the climate footprint to select tree ring records is the most consistent way to create groundwater level reconstructions for specific recharge mechanisms, despite the diverse climate and ecology found in mountain regions; and second, that modeling requires information from trees growing in at least two nearby ecoregions to create a skillful model. Within the climate footprint models, there were many more tree ring records located in the Mountain Ecoregions than in other ecoregions; however, when using the Mountain Ecoregions alone to select potential tree ring records none of the models are successful, suggesting that the addition of tree ring records from the other ecoregions help make the model successful. While using the Coast Mountain Ecoregions produces time-stable models, which have slightly higher RE and CE statistics than the climate footprint models, they explain a smaller portion of the variance in groundwater levels than the models created using the climate footprint (for example, the high-elevation recharge-driven model created using the Coast Mountain Ecoregions explains 18% of the variance (adjusted for the number of degrees of freedom), compared to 35% of the variance explained by the high-elevation recharge-driven model created using the climate footprint) (see Table 1). This suggests that while it is possible to create models using the Coast Mountain Ecoregions alone, and that this works particularly well when all the wells are considered together, including tree ring records from both the Mountain Ecoregions and the Coast Mountain Ecoregions explains more of the variance in groundwater levels than using only tree ring records from the Coast Mountain Ecoregions when the wells are divided by recharge mechanism. Therefore, using the climate footprint approach is preferable for creating strong reconstruction models based on specific recharge mechanisms because it incorporates tree ring information from both ecoregions. However, none of the methods used to select tree ring records worked for all three of the recharge mechanisms.

For the all-wells and high-elevation recharge-driven models, using the Coast Mountain Ecoregions alone created successful models, despite the smaller number of tree ring records used. This suggests that while the chronologies within the Coast Mountain Ecoregions are important for all the groundwater level reconstruction models, they are particularly important for the high-elevation recharge-driven systems. As the high-elevation recharge-driven systems are primarily recharged by direct infiltration from high-elevation precipitation, which is primarily snowmelt, groundwater levels would be expected to be correlated to tree rings in other high mountainous areas. However, the strong influence from coastal mountains suggests that the climate in the interior of BC, especially higher elevation precipitation, is related to the coastal climate. This is unsurprising given that frontal precipitation originating in the Pacific Ocean makes its way across the Coast Mountains and into the Interior [36,41]. This also corroborates the findings of Toews et al. [71], which found that in the southern BC Interior, the majority of groundwater recharge from precipitation events came from regional precipitation compared to local precipitation.

It is also interesting that when the climate footprint is used to select tree ring records, none of the records included are from the Plateau Ecoregions, even though the majority of the groundwater wells used are located with a Plateau Ecoregion. However, many of the nearby tree ring records do not extend to the year 2004, and therefore do not have enough overlap with the groundwater level records to be used in the models. In fact, there are very few tree ring records in the northern portion of the

Plateau Ecoregions with sufficient overlapping data with the groundwater levels. This is perhaps the reason why only one model could be created using the North Plateau Ecoregion.

When the three recharge mechanisms are compared (streamflow-driven, high-elevation recharge-driven, and low-elevation recharge-driven) the models created using the climate footprint use very similar tree ring records for the 1900–2004 period, which shows that these tree ring records generally correlate to changes in groundwater levels, although the relationship is weaker for low-elevation recharge-driven systems, resulting in a model that is not time-stable. However, there are some slight differences in the tree ring records used, suggesting that certain tree ring chronologies are more strongly related to one type of aquifer–stream system than others. All of the tree ring records that are used in only one of the three recharge mechanism models are high elevation chronologies (ranging from 2731 to 3000 masl). However, the chronologies that are included in only one of the three recharge mechanism models are different species for each model (Supplementary Information, Tables S5–S7). WY041 and UT535 (which were only used in the streamflow-driven model) are both limber pine, and these are the only two limber pine chronologies used for any model. WY046 (found only in the high-elevation recharge-driven model) is an Engelmann spruce chronology and is the only chronology of this species to be used in any model. Finally, MT120 (in the low-elevation recharge-driven model) is a subalpine larch, a species used in all of the reconstructions. These differences in the different chronologies found in each model suggest that some species (particularly limber pine and Engelmann spruce) may be more related to one recharge mechanism over the others.

Despite the slight differences in the species used for the reconstructions of groundwater levels in different aquifer–stream systems, the three models using the climate footprint to select tree rings preferentially used three species over the others: mountain hemlock, subalpine larch, and subalpine fir (Supplementary Information, Tables S5–S7). The chronologies selected occur at high elevations (with a minimum elevation of 1000 masl), suggesting that these high elevation tree species are more strongly correlated with groundwater levels in the interior of BC. This is likely due to the influence of snowpack on these tree species, which may limit growth by reducing the length of the growing season [11,12]. The use of multiple species has been shown to be useful for hydroclimate reconstructions because incorporating information from multiple species increases the explained variance of the reconstruction, due to small variations in how each species responds to climate e.g., [13,14,18,20,31,72–74]. Ljunqvist et al. [75] recently suggested that hydroclimate reconstructions with three or more species may be of lower quality. However, Ljungqvist et al. [75] stressed the importance of using tree ring chronologies from similar sites if multiple sites are used. The modeling process used for these reconstructions, which selects chronologies based on a significant correlation to the groundwater level records, seems to have been successful in selecting appropriate chronologies, as all of the chronologies selected for the models using the climate footprint area occur in similar environments (high elevation mountainous sites).

There are also differences in the reconstructions of groundwater levels when the models are extrapolated back into time. Despite the Coast Mountain all-wells model having the strongest RE, CE, and Pearson's r statistics during the validation period (RE: 0.496, C: 0.394, Pearson's r: 0.833), when this reconstruction is extended, it is the first to be cut off (at the year 1750) due to the RE and CE falling below zero. The Coast Mountain high-elevation recharge-driven model, on the other hand, has slightly weaker statistics during the validation period (RE: 0.349, C: 0.231, Pearson's r: 0.64), but can be extended back to the year 1500. The RE and CE statistics for this reconstruction never fall below zero, but the number of available tree ring records decreases enough that a reconstruction model is no longer possible (Supplementary Information, Figure S31). However, the last 200 years of the model are based on only one tree ring record (see Figure 6). The reconstructions created using the climate footprint have slightly lower RE, CE and Pearson's r statistics, but a higher percent variance is explained than the same models created using the Coast Mountain Ecoregions to select tree rings (adjusted r$^2$ values were 39% for the streamflow-driven reconstruction, and 35% for the high-elevation recharge-driven reconstruction). While these models are statistically valid back to the years 1700 (streamflow-driven),

1750 (all-wells), and 1500 (high-elevation recharge-driven), truncating the reconstructions at points where the RE and CE statistics drop significantly or where there are fewer than five tree ring records used to make the reconstruction model would highlight the more recent portions of the reconstructions where there is less uncertainty (shown with black arrows in Figure 6).

Based on the modeling results of this study, the reconstruction for high-elevation recharge-driven systems appears to have the most stable relationship with tree growth in the long term, which is beneficial for the purpose of extending the groundwater level record and providing insight into past groundwater variability. This could be due to the strength of the relationship to winter snowpack that is found in high-elevation recharge-driven systems, where the groundwater is directly recharged by snowmelt. The streamflow-driven reconstruction would also be expected to be related to snowpack, as the summer peak in groundwater levels is related to snowmelt, and this reconstruction could be extended back to the year 1700. The low-elevation recharge-driven systems, on the other hand, would be expected to be more related to rainfall rather than snowfall. Groundwater levels could not be reconstructed for these systems, suggesting that the relationship with snowpack is important for the groundwater level reconstructions. A closer analysis of the wells used for the low-elevation recharge-driven system average shows that only two of the wells (OW176 and OW177) are in unconfined aquifers; the remaining five wells are in deep confined aquifers, which have a lagged response compared to unconfined aquifers. This suggests that it is also important to consider aquifer type when choosing the wells to use for groundwater level reconstructions. Moreover, the ability to create reconstructions for the low-elevation recharge-driven systems may also be affected by anthropogenic influences on the watershed. When selecting groundwater observation wells, every effort was made to select wells that appeared to not be influenced by outside influences, such as pumping. Due to the low number of low-elevation recharge-driven wells, some wells with long-term declining trends had to be used, and an attempt was made to remove the effects of pumping using a detrending technique (see Section 2.2). Calibrating the models on a groundwater level series that may still have anthropogenic influences may have affected the time-stability of the low-elevation recharge-driven models. Finally, the reconstructions created using the only the Coast Mountain Ecoregions to select tree ring chronologies do not help to extend the groundwater level records further back into time, despite having higher RE, CE and Pearson's r statistics. The lower number of tree ring records used to create the Coast Mountain Ecoregion models, and the corresponding lower percent variance explained, suggests that it is preferable to include information from both the Coast Mountain and the Mountain Ecoregions to create reconstruction models when dividing the groundwater wells based on recharge mechanism. However, using the Coast Mountain Ecoregions to select tree ring records created a stronger model, with the highest RE and CE statistics, when all the wells were considered together, suggesting that there is a strong relationship between all the wells used and the snowpack signal in the Coast Mountain Ecoregions.

Although the models created using the climate footprint to select tree ring records have more model predictors than those using the smaller ecoregions to select tree ring records, the modeling process used in this study aimed to reduce the effects of overfitting in these models. In regression models, overfitting causes the model to partially describe the random noise in the data instead of the actual relationship between the variables, and therefore appear to have a strong relationship with the calibration data, but not with the validation data. The modeling process used in this study aimed to reduce overfitting in two ways: first, the number of the predictors in the regression model are reduced by using only the dominant principal components. Secondly, the Aikeke Information Criterion was used to select the final model, which evaluates the models based on their performance, and includes a penalty for more complex models. The model with the lowest AIC is therefore the model that has been determined to have an appropriate balance between the model fit and model complexity. Additionally, using cross-validation to test these models ensures that the models pass the required statistics (RE and CE) during both the calibration and validation periods.

## 5. Conclusions

In this study, records of tree ring width were used to reconstruct groundwater levels in the southern interior of British Columbia, Canada. Although tree ring records have been used to reconstruct groundwater levels previously, this is the first reconstruction of groundwater levels in a mountain region. Due to the complexities of aquifer–stream interactions, as well as the rapidly changing topography and climate in mountain areas, extra considerations were explored to determine the most robust approach for reconstructing groundwater levels. Twenty-eight groundwater level reconstruction models were created to determine (a) if the dominant recharge mechanism should be taken in account for the reconstructions, and (b) what spatial area should be used to select potential tree ring records for the reconstruction models. This analysis showed that the groundwater level reconstructions are strengthened when groundwater wells are divided into categories based on the dominant recharge mechanism (streamflow-driven, high-elevation recharge-driven, and low-elevation recharge-driven) when the climate footprint was used to select tree ring predictors, and that using a climate footprint incorporates information from tree rings in both the Coast Mountain and Mountain Ecoregions, therefore increasing the explained variance of the reconstructions. Additionally, tree ring chronologies located in the Coast Mountain Ecoregion were found to play a significant role in the models, especially in the high-elevation recharge-driven model, and created strong models with the highest RE and CE values. Extended reconstructions were made using both the climate footprint approach and the Coast Mountain Ecosystems to select tree ring records, and the longest models were created for the high-elevation recharge-driven systems. However, the uncertainty in the reconstructions increases towards the later portions of the reconstructions, with most confidence in the reconstructions occurring during the time period from 1800 to 2004 (for the streamflow-driven reconstruction) and from 1850 to 2004 (for the other three reconstructions).

Finally, one of the limitations of this study is that, currently, the period of overlap between the groundwater levels and the tree ring records is quite short (maximum 32 years for the streamflow-driven and all-wells models), and there are too few tree ring records in the 2004 to present age range to create a forward model and compare to more recent groundwater level observations. This does increase the uncertainty in the reconstructions; however, because observational groundwater level records are so short, extending these records by any amount at this time is beneficial for better characterization of these systems. New records of tree ring width are frequently added to the ITRDB, and in the future these models can be improved upon as more recent records are added and the period of overlap increases. In addition, the short period of overlap could be maximized by using a modeling procedure that allows for leave-one-out validation which, unfortunately, was not available in PcReg, the software used in this study.

**Supplementary Materials:** The following are available online at http://www.mdpi.com/2073-4433/11/12/1374/s1, Figures S1–S3: Hysteresis plots for observation well/streamflow gauge pairs; Figures S4–S7: Groundwater level records used in this study shown over the entire period of record; Table S1: List of observation wells used, with location and aquifer–stream classification; Table S2: List of tree ring sites available in the BC Interior; Figure S8–S11: Standard chronologies for the tree ring records available in the BC Interior; Table S3: Observation well/tree ring record pairs used to identify months with significant correlations; Figure S12–S23: Correlation graphs showing the months of significant correlations between the depth to groundwater level records and the tree ring chronologies; Figure S24–S29: Moving correlation plots between depth to groundwater level records and the tree ring chronologies; Figure S30: Level 2 North American Ecoregions; Table S4: Tree species considered as potential tree ring records to be used in the reconstruction models; Tables S5–S7: List of tree ring records used in the streamflow-driven, high-elevation recharge-driven, and low-elevation recharge-driven reconstruction models; Figure S31: calibration and validation statistics for extended reconstructions.

**Author Contributions:** Conceptualization, S.C.H. and D.M.A.; methodology, software, validation, formal analysis, investigation, data curation, writing—original draft preparation, S.C.H.; writing—review and editing, D.M.A., K.E.K.; supervision, D.M.A., K.E.K.; funding acquisition, D.M.A. All authors have read and agreed to the published version of the manuscript.

**Funding:** This research was funded by a Natural Sciences and Engineering Research Council of Canada Discovery Grant to D.A.

**Acknowledgments:** We thank the anonymous reviewers for their suggestions for improving the manuscript.

**Conflicts of Interest:** The authors declare no conflict of interest.

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
