# Peer review of "Comparing Approaches for Reconstructing Groundwater Levels in the Mountainous Regions of Interior British Columbia, Canada, Using Tree Ring Widths"

_atmosphere, doi:10.3390/atmos11121374_

Round 1
Reviewer 1 Report
This is the second version of the manuscript I am reviewing, and it has been substantially improved, taking into account the suggestions of both reviewers.
The manuscript can be accepted after small formatting improving.
Suggestions about figures and supplementary materials:
Figure 2 – subplot letters “(a)” and “(b)” seems too large, compared to other texts (the same for other figures with subplots). Symbols in the legend are too small to distinguish, especially for months 10 and 12. The same for Figures S1-S3
Figure 4 – in the subplots (a) and (b) there is four y-axis in different colours but the differences between those colours are too small. Maybe use more distinct colours or make new subplots.
Figure S4-S5 – very low quality. Could make those figures in other software. Remove straight lines for wells OW75, OW76 in the periods where data is missing.
Reviewer 2 Report
General comments
This study has a reasonable objective of using tree-ring data to extend groundwater level records in the southern interior of British Columbia. There are very few similar studies, and possibly none in a region of high relief and steep slopes. Unfortunately the manuscript contains extraneous and redundant information making it difficult to separate the signal (useful results) from the noise. There are several major weaknesses:
The minimal overlap of 17-32 years between the tree-ring and groundwater data is problematic as acknowledged by the authors. Their use of a split sample (2/3 – 1/3) approach to calibration / validation results in a very small size (~ 10 years) for validating the models. The authors should consider a leave one-out validation procedure that maximizes the use of short records.
The manuscript repeatedly hypothesizes that groundwater levels should correlate with tree-ring width, because both are the response to precipitation, i.e. that tree growth is limited by the availability of soil moisture. This focus on the moisture-sensitivity of tree growth might reflect the results of similar studies but in warmer and/or drier environments. The authors have an expectation of moisture sensitivity to the extent that they define the ‘climate footprint’ for selecting tree-ring data in terms of a moisture index, the SPEI. As it turned out the strongest tree-ring predictors of fluctuations in groundwater levels were ring-width chronologies from high-elevation sites where tree growth is limited by the length of the growing season. Thus there is discordance between how the authors frame their study and the results.
Section 2.1, which spans several pages, explores the statistical relationships between the ground water well observations and a relatively large amount of tree-ring data from BC, Alberta, Washington and Idaho (mapped in Figure 1). Then in the first paragraph of Section 2.2, the authors reject this analysis, noting that tree-ring data from only one site had sufficient overlap (> 30 years) with the groundwater well data. Therefore the authors proceeded to identify an expanded network of tree-ring chronologies over a much larger area. This subsequent analysis is the basis for the remainder of the paper making the first part of the study redundant.
Despite these substantive flaws, and the many minor issues identified below, this study represents a useful exploratory analysis of the potential for tree-ring reconstructions of groundwater levels in mountainous regions. The useful findings include the value added from classifying groundwater records by recharge mechanism and the common link between tree growth and groundwater levels via snowpack depth and duration. Contrary to their expectation that moisture-sensitive tree-ring data would predict groundwater levels, the authors discovered that the predictors chosen by their models were ring-width chronologies from trees growing in the vicinity of the high-elevations snowpack. Therefore, they should cite research on snowpack reconstruction from tree rings:
Spruce W. Schoenemann, Justin T. Martin, Gregory T. Pederson, David B. McWethy, (2020) 2,200-Year tree-ring and lake-sediment based for the northern Rocky Mountains highlights the historic magnitude of recent snow drought, Quaternary Science Advances, https://doi.org/10.1016/j.qsa.2020.100013.
Pederson, G. T., Gray, S. T., Woodhouse, C. A., Betancourt, J. L., Fagre, D. B., Littell, J. S., et al. (2011). The unusual nature of recent
snowpack declines in the North American Cordillera. Science, 333(6040), 332–335. https://doi.org/10.1126/science.1201570
Mood BJ, Coulthard B, Smith DJ. 2020. Three hundred years of snowpack variability in southwestern British Columbia reconstructed from tree-rings. Hydrological
Processes. 2020;1–11. https://doi.org/10.1002/hyp.13933
The results of this study also demonstrate that some tree-ring chronologies capture a regional climate signal that extends over hundreds of kilometers. With major revisions, it could be published as much more concise paper.
Specific Comments
2-3 (title): Most of the tree-ring widths were not from interior British Columbia.
58-60: This is not relevant because data from riparian trees was not used in this study.
80: Re “these studies have taken place in topographically flat areas”, earlier in this paragraph the authors refer to a reconstruction of groundwater levels using tree ring records from the Rocky Mountain foothills.
137-139: How is this approach to stratifying groundwater level records by aquifer stream system type ‘often’ used with streamflow gauges (which do not include information about groundwater)?
158: The inset map obscures tree-ring sites in northwestern Alberta. How many these sites are underneath the map and are they really ‘nearby’ to interior BC?
167-173: Yes, “moisture-limited environments are ideal for studies of tree rings”; however, most of the tree species mentioned in this paragraph grow at high elevations, where growth is temperature sensitive rather than moisture limited.
174: “may experience snow in winter” implies that occasionally there are winters without snow. If snow always occurs in winter, then this phrase is unnecessary.
179: Similar to the previous comment, this sentence suggests that precipitation is ‘possible’ in summer and therefore some summers have no rain. Every summer has some rain.
183-184: Yes, samples from moisture-limited sites are useful for reconstructing groundwater levels, but this does not describe most of the tree-ring data used in this study.
230-231: Re “annual measurements of raw ring width and monthly groundwater level records. Correlations between these two variables were analyzed”, I suspect this is an error. Ring-width data are never analyzed until they are standardized. Raw ring widths vary with the age and size of the tree.
236-238: Of the 23 tree-ring chronologies, only 5 are from tree-species that typically have moisture-limited growth. The other 18 are from high-elevation temperature-sensitive tree species.
239: Why did the authors develop chronologies for these sites, when index chronologies are available from the ITRDB?
245: Re “remove very low frequency variations, while leaving the variability due to climate intact”, some of the low-frequency variation is related to climate. Thus this statement is incorrect. What is left intact is the high-frequency variability driven by climate.
265: How close is “nearby”?
265-266: A period of overlap of 17 years is insufficient and even 32 years is marginal.
273-275: Why were only positive correlations identified? Does this assume that tree-ring width is moisture limited (which is not the case for most of the tree species used)?
277-279: Why the difference in the months ending the current spring to fall period (October versus November)?
290-294: Which 23? There are many more than 23 stands in Figure 1. If the network of tree-ring sites from “nearby” the BC Interior (shown in Figure 1) is not useful for the reconstruction of groundwater, then what was the purpose of the analyses described in section 2.1?
306-307: By using SPEI, the authors define a region of similar moisture status, implying that they are seeking tree-ring sites of moisture limited growth, but again much of the tree-ring data that they extract from the ITRDB are from high-elevation sites of temperature-sensitive tree growth.
323: How are the deserts of northern Nevada and Utah, the high plains of eastern Montana and Wyoming, and the boreal forest of northern Alberta all part of the same “climate footprint”? Use of output from the KNMI Climate Explorer requires some critical thinking and knowledge of the geography of western North America.
336-366: The methods of processing the tree-ring data were previously described in section 2.1.2. Because this study took an unnecessary two-stage approach, there is redundancy.
342: Outputting data in the ‘correct format’ is not much of an advantage. Data can always be reformatted.
346: Please refer to a study that used an ARSTAN chronology. This is very uncommon. The standard chronology retains autocorrelation and thus is generally used.
360-361: The seven regions are described below (lines 369-374) not above.
378: Re “Models were first constructed for 1900-2004”, how was that possible when the groundwater records begin only in the 1960s. A statistical model can be constructed only for the period of overlap. It would be correct to state that the initial groundwater level reconstructions were for the period 1900-2004.
385-387: Why would you pre-whiten the ARSTAN tree-ring chronologies to remove autocorrelation, when it was intentionally added to create the ARSTAN chronologies in the first place? Why not just use the residual chronologies that are produced by the ARSTAN program by pre-whitening the standard chronologies.
396-397: One-third of the overlap period would be about 10 years in most cases, that is, much too short for a reasonable validation of the tree-ring model. Consider using a leave-one-out approach to calibration / validation.
406-408: Also report the R-squared value adjusted according to degrees of freedom.
450-454: This is a good explanation of the positive correlation between tree-ring width and depth to the water table based on the persistence of a deep snowpack. Thus ring width is constrained by the length of the growing season and not the availability of moisture as implied by statements made throughout the manuscript and noted above.
490-491: Standard chronologies? On line 346, the authors stated that they used ARSTAN chronologies.
496-528: This section of the manuscript presents a lot of insignificant results that should be removed. The authors should present only their statistically significant results using the data for the Coast Mountains and larger climate footprint. There is no reason to present duplicate results (Coast and South Coast Mountains) or results from models with negative RE and CE and very low correlations.
523: The models were created using data for period of overlap which is not 1900-2004. All the reconstructions span this period but the models do not.
521-528: If “the percent variance explained is equal to the r2 value” (lines 407-408), then all of the % variance values in the Table 1 are incorrect. For example, in the upper left cell, r = 0.587 so r-squared should be 34.46% not 46.48%. Thus most of the explained variance values are very much inflated.
530-531: The first sentence of this section: “As the climate footprint was the only scenario to produce time-stable models for two of the three categories of recharge mechanism, these models were analyzed in further detail.”, should be the first sentence of the previous section. Then there would no reason to present all the meaningless results from the models other than those defined by the climate footprint.
556-558: Remove the sentence “The South Coast Mountain….”
667-668: Recalculate the % variance (see previous comment).
686: “no reconstruction was able to be made” is poor grammar. Use “we were unable to reconstruct”.
718-728: These sentences beginning with “One of the limitations …” could be moved to the Conclusions, which are sparse. Also these lines don’t follow from the preceding sentences.
